# T-Cell Infiltration and Clonality May Identify Distinct Survival Groups in Colorectal Cancer: Development and Validation of a Prognostic Model Based on The Cancer Genome Atlas (TCGA) and Clinical Proteomic Tumor Analysis Consortium (CPTAC)

**DOI:** 10.3390/cancers14235883

**Published:** 2022-11-29

**Authors:** Luca G. Campana, Wasat Mansoor, James Hill, Christian Macutkiewicz, Finlay Curran, David Donnelly, Ben Hornung, Peter Charleston, Robert Bristow, Graham M. Lord, Sara Valpione

**Affiliations:** 1Department of Surgery, Manchester University NHS Foundation Trust, Manchester M13 9WL, UK; 2Department of Medical Oncology, The Christie NHS Foundation Trust, Manchester M20 4BX, UK; 3Division of Cancer Sciences, Faculty of Biology, Medicine and Health, University of Manchester, Manchester M13 9NT, UK; 4CRUK Manchester Major Centre and Manchester Cancer Research Centre, Manchester M20 4BX, UK; 5Faculty of Biology, Medicine and Health, University of Manchester, Manchester M13 9PL, UK; 6CRUK Manchester Institute, University of Manchester, Manchester SK10 4TG, UK

**Keywords:** colorectal neoplasms, T-cell antigen receptor, prognostic factors, nomograms, prognostic model

## Abstract

**Simple summary:**

Tumour-infiltrating T-cell abundance and clonality stratify the colorectal cancer risk of death. The very highly infiltrated cancers may constitute a distinct biological subset with an immunosuppressive microenvironment despite T-cell abundance and dismal prognosis. These tumours may need personalised therapies to revert immunosuppression.

**Abstract:**

Predicting the survival outcomes of patients with colorectal cancer (CRC) remains challenging. We investigated the prognostic significance of the transcriptome and tumour-infiltrating lymphocyte T-cell receptor (TIL/Tc-TCR) repertoire and analysed TIL/Tc-TCR sequences of The Cancer Genome Atlas (TCGA) and the Clinical Proteomic Tumor Analysis Consortium (CPTAC) CRC cohorts. Using a multivariate Cox regression, we tested whether TIL/Tc-TCR repertoire, patient and tumour characteristics (stage, sidedness, total non-synonymous mutations, microsatellite instability (MSI) and transcriptional signatures) correlated with patient overall survival (OS) and designed a prognostic nomogram. A multivariate analysis (C-index = 0.75) showed that only patient age, disease stage, TIL/Tc degree of infiltration and clonality were independent prognostic factors for OS. The cut-offs for patients’ allocation to TIL/Tc abundance subgroups were determined using a strategy of maximally selected rank statistics with the *OptimalCutpoints* R package. These were “high”, “low” and “very high” (90 th percentile) TIL/Tc infiltration-stratified OS (median not reached, 67 and 44.3 months; *p* < 0.001); the results were validated in the CPTAC cohort. TIL/Tc clonality was prognostic (median OS in “high” vs. “low” clonality not reached and 67.3 months; *p* = 0.041) and independent of TIL/Tc infiltration. Whilst tumour sidedness was not prognostic, the “very highly” infiltrated tumours were prevalent among right-sided CRCs (*p* = 0.039) and showed distinct immunological features, with lower Th1 signature (*p* = 0.004), higher *PD-L1* expression (*p* < 0.001) and likely enrichment in highly suppressory IL1R1^+^ Tregs (*FoxP3* and *IL1R1* overexpression, *p* < 0.001). TIL/Tc abundance and clonality are independent prognosticators in CRC and, combined with clinical variables, refine risk stratification. We identified a subset of CRCs with “very high” TIL/Tc infiltration, poor prognosis and distinct genetic and immunologic features, which may benefit from alternative therapeutic approaches. These results need validation in prospective patient cohorts.

## 1. Introduction

An accurate, individualised outcome estimation is crucial to guide therapeutic pathways for colorectal cancer (CRC) patients [1] and the need for personalised decisions has become even more relevant following the introduction of immunotherapy and targeted agents. For example, immune checkpoint inhibitors (ICIs) have been approved only for patients with microsatellite instability [2], whereas dual anti-BRAF and anti-EGFR combinations are only available for metastatic colorectal cancers with the *BRAF* V600E mutation [3]. However, variations in patient survival and uncertainty about optimal therapeutic strategies persist for patients within the same clinicopathological staging groups [4,5].

The immune response plays a crucial role in determining the behaviour of cancers. The favourable effect of tumour-infiltrating lymphocytes (TILs) on patient outcome is well established in CRC, whose immunogenicity has been demonstrated by the tumour-selective activation of CD8+ and the migration of CD4+ T helper cells [6]. Moreover, the presence of high levels of TILs inversely correlates with signs of early metastatic invasion (vascular emboli, lymphatic invasion, perineural invasion, collectively known as “VELIPI”) and is associated with prolonged survival [7].

Although other subsets of tumour-infiltrating cells contribute to determining the patient outcome, only the combined evaluation of density and location of CD3+ and CD8+ TILs have proven to be informative for patient prognosis [8,9]. However, this approach (immunoscore) is expensive and not yet applied regularly in clinical practice or recommended for treatment selection [10].

DNA repair defects and tumour mutational burden (TMB) have been associated with better outcomes in patients treated with ICIs [11] and three neoadjuvant clinical trials (NICHE, PICC and NCT04165772) have recently reported unprecedented response rates with ICIs in patients with DNA mismatch repair-deficient (MMRd)/microsatellite instability-high (MSI-H) colon cancer [6]. Yet, a substantial proportion of MMRd tumours do not respond to ICIs, and the prognostic robustness of the TMB across cancer types has been questioned [12,13,14]. In CRC, while MMRd/MSI-H status predicts a high likelihood of response to ICIs, up to 50% of patients with MMRd/MSI-H metastatic disease will not have their disease controlled with ICIs [2], and nearly 30% of patients with non-metastatic mismatch repair-proficient (MMRp)/microsatellite stable (MSS) colon cancer may benefit from disease control with dual ICIs [15]. However, no reliable biomarkers exist to identify which MMRp/MSS cancers are the best candidates for ICIs. For these reasons, additional predictors of response are needed to select patients likely to benefit from ICIs [16].

In our previous work, we have shown that the repertoire of the tumour-infiltrating lymphocyte T cells (TIL/Tc) is a promising biomarker in melanoma and other tumours [17]. This is determined by reconstructing the sequence of the complementarity determining region 3 (CDR3) that, in mature T cells, results from the somatic rearrangement of the T-cell receptor (*TCR*) beta chain. However, in CRC, the TCR literature is heterogeneous [18,19,20,21,22,23,24,25] and without conclusive findings for patient risk stratification [22,23,25] beyond the general concept that patients with highly infiltrated (immunologically “hot”) cancers have a better prognosis. Here, to assess whether the TIL/Tc repertoire is prognostic in CRC, we used the same approach we benchmarked in other cancers [17] to characterise the T-cell repertoire and explore its correlation with tumour characteristics and patient outcome. Then, we investigated CRC samples’ mutational profile according to the TIL/Tc infiltration level. Finally, by comparing key immune microenvironment features such as Th1 activation [26] and the expression of *PD-L1* [27], *FoxP3* [28] and *IL1R1* [29], we assessed whether the different levels of TIL/Tc infiltration could be associated with diverse immunosuppressive microenvironments.

## 2. Materials and Methods

### 2.1. Data Source

We analysed The Cancer Genome Atlas (TCGA; https://www.cancer.gov/about-nci/organization/ccg/research/structural-genomics/tcga; accession number phs000178, accessed on 9 February 2018) colorectal cancer Pan-Cancer Atlas cohorts (337 cases, 75 events) using the approach we previously developed for the study of TIL/Tc repertoire [17]. The clinical data were downloaded from cBioportal (http://www.cbioportal.org, accessed on 9 February 2018) in February 2018 and merged with the updated information in 2021. All samples passed tumour purity thresholds for inclusion in the database (composed of at least 80% tumour nuclei by visual analysis http://cancergenome.nih.gov/cancersselected/biospeccriteria, as accessed on 1 April 2015). The disease stage was grouped into localised (I–II) and advanced (III–IV) according to the American Joint Committee on Cancer (AJCC) and the Union for International Cancer Control (UICC) Tumour–Node–Metastasis (TNM) staging system of malignant tumours, and tumour location divided into right-sided (from cecum to distal transverse colon) and left-sided (from splenic flexure to the rectum) CRC. Cases with missing annotations were excluded from the analysis. We utilized the total non-synonymous mutation count and the Microsatellite Analysis for Normal-Tumor InStability (MANTIS) microsatellite instability (MSI) score (which was calculated by TCGA using the MANTIS program that is reported to have a diagnostic accuracy of 98.9%, pipelines described in Bonneville et al. [30] and Kautto et al. [31]; the higher the score, the higher the likelihood of microsatellite instability, Appendix A Appendix A) data as available in the TCGA official clinical metafile. The *IFN-γ*, *PD-L1*, *FoxP3* and *IL1R1* mRNA z-scores were obtained from the cBioportal “gene query” data download page (batch corrected). We used the xCell scores of the TCGA colorectal cancer cohort [32] to deconvolute the Th1 signature, as publicly available from xCell portal. We downloaded the masked somatic mutation data (varscan somatic MAF file) from dbGap for TCGA colorectal cancer series (phs000178 COAD and READ cohorts) and used the R package “*maftools*” to analyse the single-nucleotide variations (SNVs), the single-base substitutions (SBSs) signatures, the differentially mutated genes and visualized the data. We downloaded the BAM files of the bulk RNA-Seq and the clinical metafile of the validation cohort (8 events in 106 cases) from the Clinical Proteomic Tumor Analysis Consortium (CPTAC-2) repository colorectal cancer dataset (https://portal.gdc.cancer.gov/projects/CPTAC-2, accession number phs000892 study ID “coad_cptac_2019”, accessed on 14 August 2021) [33].

### 2.2. T-Cell Receptor Analysis

T cells uniquely undergo somatic rearrangements of their *TCR* locus in the CDR3; as a result, the mRNA sequences of the not-native, rearranged *TCR* genes are unique to T cells. The transcription of TCR mRNA is stable overall in T cells, and thus the *TCR* mRNA quantification can be considered indicative of the number of T-cell genomes present in the sample that was sequenced. Due to the very high variability of the random rearrangements of the *TCR* gene, the presence of identical *TCR* genes likely indicates that the sequences come from the same T-cell expanded clonotype [24,34]. CDR3 *TCR* sequence data for the CRC–TCGA cohorts were made available by the authors of the original publication (Appendix A Appendix A) [24] as per our previous work [17]. The productive *TCR* sequences of the CPTAC–CRC cohort were reconstructed with *MiXCR* (v.3.0.12) [32] from the FASTQ files generated after indexing (samtools v.1.9) of the RNA-Seq BAM files. The inclusion criterion for the *TCR* downstream analyses was a minimum of 4 *TCR* sequences identified. *TCR* diversity was calculated using the Renyi index with alpha = 1 (an ecological method well benchmarked to the study of T-cell clonotype distribution; the higher the value, the higher the diversity, with a range from 0 to 1) as per the algorithm utilised in Spreafico et al. [35]. Clonality was calculated as the inverted normalised Shannon entropy, which is the ratio between the frequencies of all productive sequences and the logarithm of the total number of unique productive sequences (the higher the value, the higher the clonality, with a range from 0 to 1), using the function “*clonality*” of the “*lymphoseq*” R package (further details of the approach and pipelines described elsewhere [35,36,37]).

### 2.3. Statistical Analyses

Within the limitations caused by the incomplete annotation of the retrospective series, this study follows the reporting recommendations for tumour marker prognostic studies (REMARK) guidelines by providing transparent information on patient characteristics, specimen selection, marker assay methodology and statistical analysis [38]. Tests were two-sided, with *p* values < 0.05 considered significant. Multivariate Cox regression was used to calculate the hazard of death; we applied the fast-backwards method (*fastbw* function in *rms* R package, *type = “individual”*) to assess the joint predictive ability of covariates. Selection starts with all potential explanatory covariates included in the model and eliminates the least important covariates early on, leaving only the most important variables in the model. To design the final model, Akaike Information Criterion (AIC) was used as a stopping rule to weight the probability of both significance and prediction strength. The model performance was quantified with C-index after bootstrap validation (“*validate*” in rms R package, 200 bootstraps, rule = “*aic*”) and compared with analysis of variance. Due to the limited number of events and the data distribution in the CPTAC cohort, we could not validate our analysis for TIL/Tc clonality in that series. Cox–Snell residuals were used to verify the proportional hazard hypothesis (*p* value > 0.05 confirmed the proportionality for all covariates retained in the final model). Comparisons were made with Chi-square or Fisher’s exact test (according to group sizes; categorical variables) and Mann–Whitney test or Kruskal–Wallis analysis of variance test (according to the number of groups; numerical variables). Kaplan–Meier method and log-rank test were used to compare survival curves. Using Cox regression modelling, we determined that TIL/Tc clonality conferred a continuous linear decrement of the hazard of death, while TIL/Tc abundance had a discrete relationship with the hazard of death (two splines). Thus, we categorised TIL/Tc clonality and abundance in the Kaplan–Meyer curves and TIL/Tc abundance in the prognostic study, with the cut-offs for patients’ allocation to groups determined using a strategy of maximally selected rank statistics [39] with the OptimalCutpoints R package [40], a tool developed to select optimal cut points for diagnostic tests (as per methodology applied in our previous work [17]). Analyses were performed with GraphPad Prism version 7 (GraphPad Software, La Jolla, CA, USA) or R (v. 3.6.3, The R Foundation for Statistical Computing, Vienna, Austria).

## 3. Results

### 3.1. Study Population

The patient characteristics are listed in Table 1. The study population was balanced regarding primary tumour anatomic location (right colon, 47%; left colon, 53%) and disease stage (we grouped the stages into two balanced groups: I–II, 54%; III–IV, 46%). All patients except one were treatment-naive.

### 3.2. Prognostic Analysis

To determine whether the TIL/Tc repertoire metrics correlated with patient overall survival (OS), we performed a Cox regression multivariate analysis including standard clinical prognostic factors (age, stage, sidedness), TIL/Tc infiltration, clonality and diversity, SNV signature, MANTIS–MSI score and *IFN-γ* mRNA z-score. First, to simplify the classification, we estimated the fittest cut-off points for TIL/Tc infiltration (total number of T-cell genomes sequenced in the bulk RNA-Seq) and TIL/Tc clonality (inverted Shannon entropy) using maximally selected rank statistics [39], following the methodology we applied in our previous works [17,41]. We then proceeded to variable selection using a backwards selection approach. As expected, older age and advanced disease conferred a significantly worse prognosis. Furthermore, we discovered that a high TIL/Tc degree of infiltration and high clonality were independent prognostic factors associated with longer OS. Intriguingly, we also found that the subset of patients with cancers in the top ten percentiles of TIL/Tc infiltration (“very high” infiltration) had a worse hazard for mortality than the other patients (Figure 1a), whereas the remaining variables were not significant, including primary tumour sidedness (Appendix A Appendix A). Similarly, the total SNV was not retained as significant, in line with the observations that tumour mutational burden may be of secondary importance compared to immunological factors [13].

Finally, we built a prognostic model using the selected variables and proceeded with internal validation using the bootstrap strategy (200 bootstraps) with a C-index = 0.75. To facilitate the estimation of the risk of death based on the prognostic model, we developed a nomogram (Figure 1b). An example of its application is provided in the Appendix A Appendix A, where we estimated two- and five-year OS probabilities for a hypothetical patient.

### 3.3. Survival Analysis

Consistent with our regression model, we found that the patients with high-TIL/Tc clonality tumours lived longer (median OS not reached) than those with low-TIL/Tc clonality tumours (median OS 67.3 months, *p* = 0.041; Figure 1c).

To assess whether the TIL/Tc degree of infiltration and clonality could be linked, we measured TIL/Tc clonality in the three identified TIL/Tc infiltration groups and found no significant correlation (median TIL/Tc clonality = 0.23, 0.20 and 0.18 in the “low”, “high” and “very high” TIL/Tc infiltration groups, respectively; *p* = 0.2042, Appendix A Appendix A)

Finally, we did not observe any relevant differences in TIL/Tc clonality between the right-sided vs. left-sided (median 0.21 vs. 0.20, respectively; *p* = 0.673, Appendix A Appendix A), and stage I–II vs. III–IV (median 0.20 vs. 0.23, respectively; *p* = 0.438, Appendix A Appendix A) tumours.

We then focused on analysing the total TIL/Tc degree of infiltration. We identified three distinct prognostic groups, where patients with “high” TIL/Tc infiltration had the better prognosis (median OS not reached), followed by those with “low” TIL/Tc infiltration (median OS of 67 months) and those with “very high” (i.e., above the 90th percentile) TIL/Tc infiltration, which had the shortest OS (median, 44.3 months, *p <* 0.001; Figure 1d).

Since this finding was partly unexpected, we validated this observation using the CPTAC colorectal cancer cohort (*n* = 106) [33] and confirmed that the individuals with “very highly” infiltrated cancers (>90th percentile) survived for a shorter time than the other patients (*p* = 0.005, Appendix A Appendix A).

### 3.4. TIL/Tc Degree of Infiltration According to Clinical and Molecular Characteristics

#### 3.4.1. Disease Stage

To test whether TIL/Tc infiltration could be influenced by disease stage, we compared the degree of TIL/Tc infiltration in stage I-II and III-IV tumours and found no differences (10% vs. 11%, respectively; *p* = 0.708, Appendix A Appendix A).

#### 3.4.2. CRC Sidedness

Despite being associated with microsatellite instability and more aggressive clinical behaviour [42], having a right-sided colon tumour did not affect prognosis in our multivariate analysis. To further investigate this finding, we first compared the total non-synonymous mutations, as available in TCGA clinical metafiles, with primary tumour sidedness and, not unexpectedly, found that the right-sided CRCs had more total non-synonymous mutations than the left-sided ones (*p* < 0.001, Appendix A Appendix A). This is consistent with the higher incidence of microsatellite instability (MSI) in the right-sided CRCs [43].

To investigate whether this was associated with a different mutagenesis signature, we then analysed the pattern of mutations and found that both right- and left-sided CRCs had prevalent SNV dominated by C > T (Appendix A Appendix A), as commonly seen in the age-related signature 1 dominance [44]. Conversely, right-sided cancers had a relative enrichment in T > G as the second dominant SNV (*p* = 0.0026, Appendix A Appendix A), arguably related to defects in the DNA repair pathway [44]. Consistent with this, we showed that the MANTIS–MSI score, used to predict the MSI status (the higher the value, the higher probability the tumour is microsatellite unstable) [30], was significantly higher in right-sided tumours (median: 0.34 vs. 0.36 in left- and right-side cancers, respectively; *p* < 0.001) (Appendix A Appendix A). However, the MANTIS–MSI score was not prognostic for survival neither in the entire cohort (Appendix A Appendix A), nor in the subset of CRC cancers with “very high” TIL/Tc infiltration (Appendix A Appendix A).

Finally, we tested whether there was a difference in TIL/Tc infiltration according to tumour sidedness and found a preponderance of right-sided vs. left-sided CRCs in the subset of patients with “very highly” infiltrated tumours (66% vs. 34%, respectively, *p* = 0.039). In other words, among the right-sided CRCs, there is an excess of “very highly” infiltrated tumours compared to the left-sided ones (14% vs. 7%, respectively, *p* = 0.039, Figure 2).

#### 3.4.3. SNV Pattern

Since TIL/Tc infiltration could be related to neoantigen burden and since we observed an excess of “very highly” infiltrated tumours in the right-sided CRCs (which also had higher SNV values and MANTIS–MSI score), we tested the correlation between the TIL/Tc and SNV pattern. We found that the total non-synonymous mutation count was similar across the three TIL/Tc infiltration groups (*p* = 0.517, Figure 3a), supporting the hypothesis that TIL/Tc infiltration is an independent prognostic factor and not a direct correlate of the TMB.

Similarly, and in contrast to the differences found between right- and left-sided CRC, we did not find a significant difference in the second dominant SNV distribution between the “very high” infiltrated cancers and the others (the “low” and “high” TIL/Tc infiltration groups were merged to allow for comparisons; *p* = 0.414, Figure 3b).

We then expanded the analysis of the SNV in the context of the ‘3 and 5′ flanking nucleotides to calculate the SBS tri-nucleotide signatures [45] and found that the three best matches were SBS Signature 10a/b (polymerase epsilon mutations), Signature 1 (deamination of 5-methylcytosine, usually associated with ageing) and Signature 6 (defective DNA mismatch repair) for both cohorts (Appendix A Appendix A). Furthermore, even the MANTIS–MSI score was not significantly different across the TIL/Tc infiltration groups (*p* = 0.089, Figure 3c), again supporting the hypothesis that TIL/Tc infiltration is not directly dependent on the TMB or the mutation aetiology.

### 3.5. Characterisation of the “Very Highly” Infiltrated CRCs

#### Mutational Profile

The total SNV, nucleotide variation profile and MANTIS–MSI analysis did not identify a different mutational process associated with “very high” infiltration. Thus, we next wanted to determine whether the condition of “very high” infiltration was associated with genetic differences. Hence, we compared the most common mutations in the “not very highly” (i.e., “low” plus “high”) vs. “very highly” infiltrated cancers (Figure 4). As expected, the former displayed a mutational profile consistent with the ubiquitous signatures of CRC, led by *APC* and *TP53* (mutated, respectively, in 72% and 61% of patients, Figure 4a) [46]. Conversely, in the “very highly” infiltrated subset, the predominant mutation was *TTN* (63% of patients), whereas *APC* and *TP53* were mutated only in 47% of patients (Figure 4b). Additionally, *KRAS* was the fourth most commonly mutated gene in the “not very highly” infiltrated CRCs (41% of patients), whereas it did not appear among the top twenty genes in the “very highly” infiltrated group, having an incidence of <30%. Conversely, *BRAF* was not in the top twenty in the “not very highly” infiltrated CRCs (<15% of cases). In contrast, it ranked sixth among the most common mutated genes in the “very highly” infiltrated group (37% of patients) and was also among the genes with a significant odds ratio of mutations in the “very highly” infiltrated CRCs (Appendix A Appendix A). Intriguingly, the majority of the “very highly infiltrated” CRCs with a BRAF mutation were stage I or IIA (8/11), whilst only one case was stage IIIC and none was stage IV in this subset (Appendix A Appendix A). Consequently, most of this patient subset would fall into the good prognostic group based on the clinical AJCC classification. Collectively, our findings show that the “very highly” infiltrated CRCs display distinct genetic features.

### 3.6. Immune Microenvironment

“Very highly” infiltrated CRCs are generally considered immunogenic “hot” tumours. Therefore, we hypothesised that an immunosuppressive immune microenvironment may explain the counterintuitive dismal prognosis in these patients. First, we used xCell [32] to assess the Th1 signature, which characterises efficient anticancer cytotoxicity, and found a significant defect in the “very highly” infiltrated tumours (*p* = 0.004; Figure 5a).

Moreover, in the same subset, we detected an excess of *PD-L1* expression (*p* < 0.0001, Figure 5b). Finally, we found indirect evidence of an excess of the highly suppressive IL1R1^+^ Treg subset [29] in “highly infiltrated” CRCs since they had overexpression of *FOXP3* (*p* < 0.001, Figure 5c) and *IL1R1* (*p* < 0.001, Figure 5d) mRNA.

Taken together, these results show that the “very highly” infiltrated CRCs are a distinct biological subset with features of dysfunctional tumour immune infiltrate.

## 4. Discussion

Approximately one in five patients will develop disease recurrence after definitive surgical treatment of CRC due to the presence of micrometastatic disease at the time of intervention [47]. Perioperative neo-/adjuvant systemic treatment may eradicate microscopic residual disease and improve patients’ outcomes [15,48,49,50]; however, despite the recent development of accurate prognostic models based on patient, tumour and treatment-related risk factors [47], sole clinicopathological information is suboptimal in predicting CRC patients’ outcome and guiding therapeutic choices; therefore, ongoing research efforts are focused on identifying these high-risk patients [10,51,52]. To address this unmet need, we extended our previous work on immune biomarkers [17] investigating the prognostic significance of the TIL/Tc repertoire in two independent CRC cohorts.

Although the findings of this study need prospective confirmation and our prognostic model does not lend itself to immediate clinical application, our results are hypothesis-generating since they can pave the way for better patient stratification and provide biological insights for future tailored treatments. First, we demonstrated that the TIL/Tc repertoire identifies patient subgroups with distinct prognoses and, specifically, the abundance and clonality of TIL/Tc were independent prognostic factors and made it possible to stratify patients into subgroups with different OS. Second, we identified a subset of tumours with “very high” TIL/Tc infiltration, dismal prognoses and distinct clinical, immunologic and genetic features, which have not been previously described.

### 4.1. Prognostic Role of TIL/Tc Clonality

The significant prognostic role of TIL/Tc clonality is corroborated by the observation of a reduction in *TCR* diversity (inversely correlated with clonality) in patients achieving the most durable response from immunochemotherapy with peptide vaccines and oxaliplatin [22] and also, more recently, by a phase II trial where the pre-treatment of TIL/Tc clonality identified different likelihoods of response to ICIs [53]. Notably, TIL/Tc clonality is not a biomarker for minimal residual disease since it did not correlate with better recurrence-free survival after radical surgery in the analysis of 575 patients with primary stage II CRC [25].

### 4.2. Prognostic Role of TIL/Tc Infiltration

The assessment of TIL/Tc infiltration has long been used to improve outcome prediction in CRC patients, and most prognostic tools rely on immunohistochemistry-based quantification [54]. However, since this approach considers TIL/Tc as a continuous variable incrementally correlating with a favourable outcome with only a few exceptions [55], it has failed to identify the small subset of CRC with very high TIL/Tc infiltration. Intriguingly, in the TCGA cohort, the median OS of the subgroup of patients with “highly infiltrated” CRCs is only 44 months, strikingly similar to those with inflammatory breast cancer [56]. The ultimate effect of the antitumour immune response depends on several factors, including, among others, its quantitative/qualitative composition (e.g., the balance between activated, exhausted or regulatory T cells) and localisation within the microenvironment [57]. Thus, somewhat paradoxically, chronic inflammation and abundant TIL/Tc infiltration can be a double-edged sword and support tumour progression by promoting angiogenesis, proliferation and metastatisation [58,59,60].

### 4.3. Immunologic Features of the “Highly Infiltrated” CRCs

We demonstrated that the “highly infiltrated” CRCs have a defect in Th1 cytotoxic signature, an excess of inhibitory checkpoint molecule PD-L1 and are likely rich in IL1R1^+^ Tregs, a subset of cancer-specific and high immunosuppressive Tregs. Notably, the “very highly” infiltrated tumours were not restricted to MSI CRCs, which currently represent the only approved indication for treatment with ICIs. Conversely, the “very highly” infiltrated cancers were likely enriched in PD-L1, the standard-of-care eligibility biomarker for ICIs treatment in several malignancies, including non-MMRd/MSI upper gastrointestinal cancers. Taken together, these considerations encourage future testing of ICI, including at early stages, in CRC with very high TIL/Tc infiltration. In these circumstances, an immune-suppressive microenvironment may benefit from therapies directed to reduce checkpoint inhibition or to clear IL1R1^+^ Tregs (e.g., with novel anti-CD25 therapies [61]) and where the elevated risk of death for these patients would balance the potential toxicity from therapy in the neo-/adjuvant setting. Of relevance, 4-week neoadjuvant checkpoint inhibition with a single dose of ipilimumab and two doses of nivolumab has been shown to achieve deep or complete pathologic responses in patients with either MMRd and MMR-proficient early stage colon cancers, thus providing evidence for potentially practice-changing paradigm shifts [15].

### 4.4. Clinical Features of the “Highly Infiltrated” CRCs

The “very highly” infiltrated CRCs were predominant in the right colon. Historically, right-sided CRCs have been described as having differing embryology, distinct molecular features, diverse microbiota and a worse prognosis than left-sided tumours [62]; however, when analysed in multivariate regression, CRC sidedness was not significant, but TIL/Tc infiltration was. Our observations suggest that historically, the dismal prognosis in right-sided CRC may reflect, at least in part, an enrichment in “very highly” infiltrated tumours. Additionally, our findings may contribute to explaining the better responses to immunotherapy observed in right-sided tumours.

### 4.5. Genetic Features of the “Highly Infiltrated” CRCs

Compared to the other CRCs, in addition to a distinct immunological landscape, the “very highly” infiltrated subset was distinguished by a higher relative incidence of *BRAF* mutations (Figure 4, Appendix A Appendix A).

Notably, most *BRAF*-mutated cancers with “very high” TIL/Tc infiltration in the TCGA cohort had stage I or IIA, only one patient had stage IIIC, and none was metastatic, so the hypothesis that an advanced clinical stage could drive the worst prognosis is unlikely. Intriguingly, several trials are underway combining checkpoint inhibition and BRAF-targeted therapy for *BRAF*-mutated CRC (ClinicalTrials.gov (accessed on 25 November 2022) identifier: NCT03668431, NCT04017650, NCT04294160) because the majority of *BRAF*-mutated CRCs are Consensus Molecular Subtype 1 (immunogenic and hypermutated) [63], again supporting our hypothesis that the genetic background and the tumour microenvironment of CRC are correlated and identify a separate entity that may benefit from personalised therapy approaches, possibly combining chemotherapy and targeted drugs with immunotherapy.

### 4.6. Study Limitations

The results of this study are hypothesis-generating and should be interpreted in light of several limitations inherent to the nature of the data. First, we had no access to detailed information on primary tumour characteristics and patient clinical trajectory (e.g., treatments, time to first recurrence, survival after recurrence and MSI assessment with a clinically validated method). Moreover, the relatively small number of patients prevented a more granular stage subgrouping and a more robust assessment of the prognostic value of TIL/Tc infiltration within each AJCC staging group. Although using the 90th percentile cut-off to define the “very high” TIL/Tc infiltration subgroup (as determined by the strategy of maximally selected rank statistics that we applied), our analyses captured a subset of CRCs with dismal prognosis in two independent cohorts. At the same time, we acknowledge that cut-off strategies are intrinsically dataset- and methodology-dependent, so future studies are needed to standardise the *TCR* repertoire estimation in a scalable platform that could be benchmarked for clinical application.

Additionally, we could not differentiate between tumour-infiltrating and peritumoural T cells or perform a differential analysis of the T-cell signatures that can only be achieved with single-cell technologies. Similarly, we could not correlate immunohistochemistry and RNA-Seq data to compare the accuracy of our model with the Immunoscore^®^ assay, so we warrant prospective comparisons. A comprehensive study of TIL/Tc infiltration abundance and geography, neoantigen specificity, clonal expansion and phenotypic subsets in the tumour niche is paramount to advancing our understanding of the mechanisms that drive tumour immune escape mechanisms. Additionally, due to the lack of detailed information about patient body mass index and systemic treatment in the cohorts we investigated, it was impossible to investigate any relationship between obesity and tumour infiltration [64] or determine the impact of different therapies on survival [38]. Therefore, we endorse validation in prospective series, particularly in the ICIs scenario, also using targeted *TCR* sequencing tools that, by increasing the detection power for *TCR* sequences [17], can facilitate the standardisation of optimal cut-offs and the qualification of *TCR*-based biomarkers. Although the pointwise analysis of neoantigen specificity and functional status of the expanded TIL/Tc clonotype and the “highly infiltrating pools” was beyond the bandwidth of the present study, our results nonetheless warrant further validation of TIL/Tc clonality and degree of infiltration.

## 5. Conclusions

This study contributes to addressing the unmet clinical need for precise and reliable prognostic biomarkers in CRC by identifying distinct risk groups based on the TIL/Tc-*TCR* repertoire. We also identified a subset of CRCs, not previously described, characterised by very high TIL/Tc infiltrate and distinct clinical (prevalence in the right colon), genetic (e.g., *BRAF* mutation) and immunological (low Th-1 signature, high *PD-L1*, *FOXP3* and *IL1R1* expression) characteristics, suggesting a framework for personalised therapeutic strategies. We recommend expanding these findings to prospective patient cohorts to validate these results and optimise scalable approaches for clinical qualification.

## Figures and Tables

**Figure 1 cancers-14-05883-f001:**
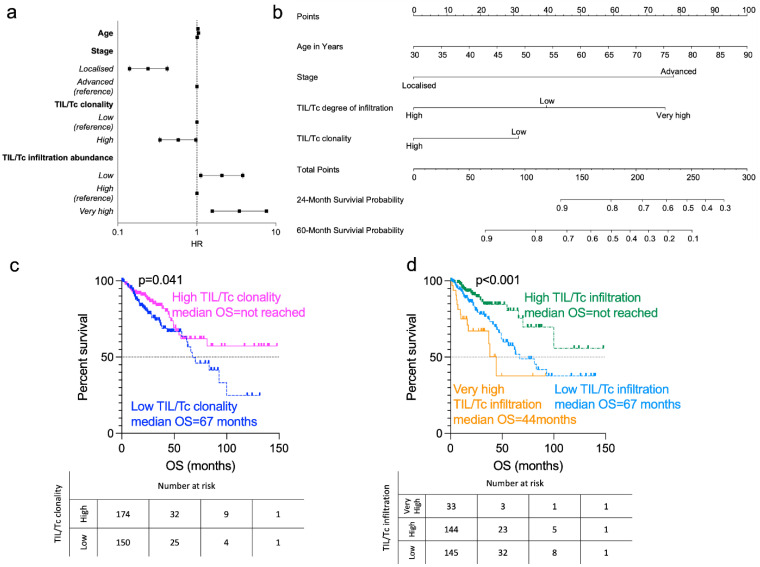
Prognostic factors for overall survival in patients from The Cancer Genome Atlas colorectal cancer cohort. (**a**) Forest plot showing hazard ratio (HR) for mortality with 95% C.I. for significant covariates retained in the multivariable Cox regression prognostic model calculated using the fast–backwards method and the Akaike information criterion as a stopping rule and summarised in Appendix A Appendix A (cases = 337, Cox model for OS global *p* < 0.001). Age, HR 0.03, *p* < 0.0001; stage (localised), HR 0.34, *p* < 0.0001; TIL/Tc clonality (high), HR 0.86, *p* = 0.002; TIL/Tc infiltration (low), HR 0.55, *p* = 0.011; TIL/Tc infiltration (very high), HR 1.43, *p* < 0.001. (**b**) Derived nomogram for survival probability integrating clinical variables and TIL/Tc features. (**c**) Survival curves for patients with high (pink) or low (blue) TIL/Tc clonality (log-rank *p* = 0.041). (**d**) Survival curves for patients with “low” (blue), “high” (green) or “very high” (orange) TIL/Tc abundance (log-rank *p* < 0.001).

**Figure 2 cancers-14-05883-f002:**
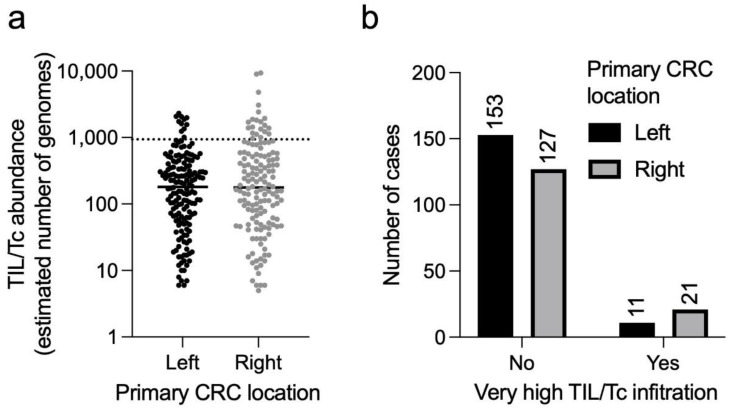
TIL/Tc infiltration according to CRC sidedness. (**a**) Estimated number of T cell genomes in the sample, indicative of TIL/Tc abundance, according to the cancer site. The dotted horizontal line represents the threshold for very high TIL/Tc abundance (90th percentile). (**b**) The same cases are grouped according to anatomical location and TIL/Tc abundance (“low or high” = below the threshold indicated in panel (**a**); “very high” = above the same threshold. Fisher’s exact test *p* = 0.039). Horizontal bars in (**a**) represent median values.

**Figure 3 cancers-14-05883-f003:**
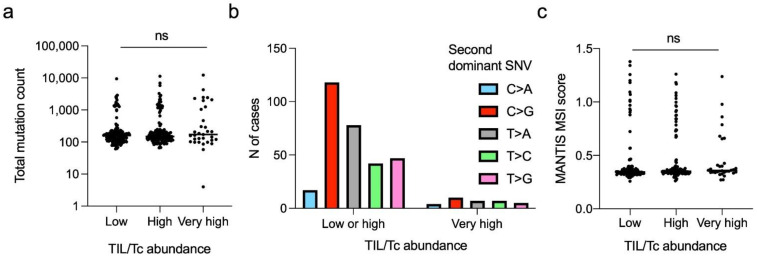
Relationship between TIL/Tc abundance and mutational landscape. (**a**) Total non-synonymous mutation count in cancers with low (median = 159), high (median = 119) and very high (median = 108) TIL/Tc abundance (Kruskal–Wallis analysis of variance test *p* = 0.5170). (**b**) Distribution of the second dominant SNV in cases with low/high and very high TIL/Tc abundance; relative to total SNV, the prevalence of C > A was 6% vs. 12%, the prevalence of C > G was 39 vs. 30%; the prevalence of T > A was 26 vs. 21%; the prevalence of T > C was 14 vs. 21% and the prevalence of T > C was 16 vs. 15% (Chi-square test *p* = 0.4146). (**c**) MANTIS–MSI scores in cancers with low (median = 0.35), high (median = 0.35) and very high (median = 0.26) TIL/Tc abundance (Kruskal–Wallis analysis of bariance test *p* = 0.0893). Horizontal bars in (**a**,**c**) represent median values. ns, not significant.

**Figure 4 cancers-14-05883-f004:**
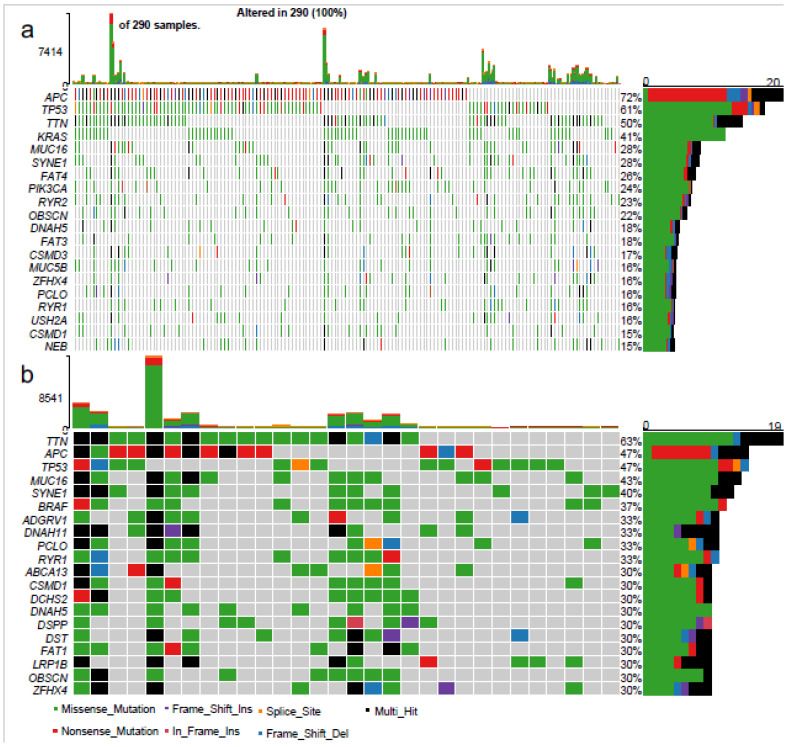
Oncoplot of the 20 most frequent mutations in “not highly infiltrated” vs. “very highly infiltrated” CRCs. (**a**) Top mutated genes in the “not highly infiltrated” (“low” plus “highly infiltrated) CRCs. (**b**) Top mutated genes in the “very highly” infiltrated CRCs. The upper chart of the diagrams represents the number of hits for every single patient. On the left are the genes’ names; the grid represents the gene status for every patient (the colour code for the alteration subtypes is in the legend on the bottom); on the right, the percentual frequency and the absolute number of each gene alteration.

**Figure 5 cancers-14-05883-f005:**
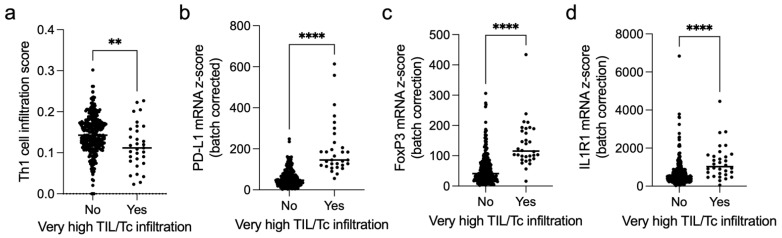
Unfavourable tumour microenvironment in the “very highly” infiltrated CRCs in the TCGA cohort. (**a**) Th1 infiltration score as measured by xCell [33] in the “low/highly infiltrated” vs. “very highly infiltrated” tumours (median 0.14 vs. 0.11, respectively, *p* = 0.004). (**b**) *PD-L1* expression in the “low/highly infiltrated” vs. “very highly infiltrated” tumours (median z-score 47.3 vs. 145.7, respectively, *p* < 0.001). (**c**) *FOXP3* expression in the “low/highly infiltrated” vs. “very highly infiltrated” tumours (median 41.32 vs. 115.70, respectively, *p* < 0.001). (**d**) *IL1R1* expression in the “low/highly infiltrated” vs. “very highly infiltrated” tumours (median 510.3 vs. 1026, respectively, *p* < 0.001). ** *p* <0.01; **** *p* < 0.0001.

**Table 1 cancers-14-05883-t001:** Patient characteristics in TCGA cohort (*N* = 337).

Variables	*N*. or Median (%/Range)
**Age at diagnosis**	67 (33–90)
**Sex**	
Female	154 (47)
Male	175 (53)
n.a.	*8*
**Disease stage**	
Localised	168 (54)
I	52 (17)
II	116 (37)
Advanced	143 (46)
III	100 (32)
IV	43 (14)
n.a.	*26*
**Primary tumour side**	
right	148 (47)
left	166 (53)
n.a.	*23*
**Neoadjuvant therapy**	
no	328 (99.7)
yes	1 (0.3)
n.a.	*8*
**Total TIL/Tc genomes ^a^**	182 (5–9336)
**TIL/Tc clonality**	0.21 (0.03–0.76)
**TIL/Tc diversity**	2.7 (0.2–5.3)
**Total SNVs**	155 (4–12338)
**MANTIS–MSI score**	0.35 (0.26–1.38)
**IFNγ mRNA z-score**	−0.24 (−1.76–3.34)

Note: TIL/Tc = tumour-infiltrating lymphocytes T cells (as calculated from RNA-Seq TCGA data); MANTIS = microsatellite analysis for normal-tumor instability; MSI = microsatellite instability; SNV = single-nucleotide variation; *N* = number of samples; n.a. = not available. ^a^ TIL/Tc infiltration.

## Data Availability

The script for the maftool analyses and the complete Prism file with the source data for the paper figures and the analyses is available from the repository https://github.com/SaraValpione/CRC_Tregs (accessed on 28 November 2022). Other material (metafile with the *TCR* metrics) is available upon reasonable request from the corresponding authors.

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
