# Peer review of "T-Cell Infiltration and Clonality May Identify Distinct Survival Groups in Colorectal Cancer: Development and Validation of a Prognostic Model Based on The Cancer Genome Atlas (TCGA) and Clinical Proteomic Tumor Analysis Consortium (CPTAC)"

_cancers, 2022, doi:10.3390/cancers14235883_

Round 1

Reviewer 1 Report (Previous Reviewer 2)

The authors have improved the manuscript, although the language could be improved, e.g. "provocative" can be changed, and the introduction is still too concise, however, it is scientifically sound.

Author Response

Please see the attached Word file.

Reviewer 2 Report (New Reviewer)

In the revised manuscript the authors have satisfactorily addressed all reviewers’ questions and concerns.

In particular, the mention of the limitations inherent to the data now clearly mitigates the findings reported in the manuscript highlighting the exploratory and hypothesis-generating nature of the study thus avoiding misinterpretation bias.

Moreover, the new data and Figures proposed in the revised manuscript, as well as the addition of specific information and details in “Materials and Methods” section, have improved the clarity and interpretation of findings reported. 

Comment 1

The authors have identified a small subset of “very highly” infiltrated CRC (N=~30) with dismal prognosis, distinct clinical, immunologic and genetic features.

This subset is an heterogenous population composed by 1:1 advanced (stage III-IV) and localized (stage I-II) cases (Supp Figure 6) with preponderance of right-sided CRC (Figure 2). Moreover, the mutation profiling of the “very highly” infiltrated group highlights a higher incidence of BRAF mutations compared to the other CRCs. As reported in Figure 4b, BRAF mutations seems to be associated to high TMB (8 cases out 10 with high-TMB) and thus likely MSI tumors. Are these cases also advanced (stage IV) tumors? If yes, taking in consideration the limited number of cases in this group, this potential bias could largely affect the OS analysis reflecting the worse prognosis of BRAF mutated mCRC. Moreover, are there some differences in terms of prognosis in MSS vs MSI “very highly” infiltrated CRC? The authors should comment on this to better characterize the biological and clinical features of the “very highly” infiltrated CRC.

Comment 2

In the manuscript is not reported the numerosity of each subgroup of patients identified (Figure 1c and 1b). To better clarify this aspect, the authors should add below the survival curves the number of patients at risk in each subgroup. Similarly, this information should be added also in Supplementary Figure 5.

Author Response

Please see the Word file.

This manuscript is a resubmission of an earlier submission. The following is a list of the peer review reports and author responses from that submission.

Round 1

Reviewer 1 Report

This is a potentially interesting but somewhat untidily written manuscript reporting the results of a computational analysis of two multimodal genomic cancers databases - TGCA and CPTAC -, suggesting that TIL abundance and clonality, as expected, are prognostic markers for survival in colorectal cancer (CRC). Unexpectedly however the authors find that very highly infiltrated CRC have a dismal prognosis due to an immuno-suppressive microenvironment despite T cell abundance. 

Major issues

1.     General: findings suffer from a possible misclassification bias due to insufficiently characterized clinical landscape of the cohort and a consequent reduced choice of variables for the analysis. 

This is somewhat intrinsic of TGCA based analysis, but the potential conclusion-distorting consequences are not fully recognized by the authors.  To say that the population was well balanced by site (L vs R) without knowing whether the primary tumor was in the colon or the rectum might be misleading since the MSI and HER2 phenotypes are abundant in rectal cancer (less so in the sigma or descending colon). Even more misleading is the definition of localised (I-II) or an advanced (III-IV) stage disease rather than using the TNM classifier. As an example, as cited in ref 36 the lymph node index carries a survival hazard ratio (HR) of 5.02 (95% CI 3.94–6.40) compared to a mere 1.1 for age. This dichotomy (I/II vs III/IV) is clinically and biologically inaccurate since: 1) survival is extremely different for frankly metastatic stage IV while at least 50% of N+  cases (Stage III)  will NOT develop metastases and presumably die of non cancerous causes; on the contrary up to 20% of early stage (including stage I) are already metastatic at the time of surgery, will become frankly metastatic within 2 years (especially T4N0) and die of their cancer. The absence of information about treatment could also introduce further biases since treatment rescues about 30% of potentially (immune?) ‘bad’ tumors. One wonders if these potential biases are responsible for the unexpected, counterintuitive and unexplained result of the ‘very high TIL/Tc’ subgroup doing much worse than the low and high subgroups (fig 1d). The author reason that the finding is consistent with findings in a second validation cohort (CPTAC cases, fig S3), but again the true biological meaning could have been shadowed by the same potential misclassification bias. Ditto for the lack of difference by disease stage, regarding the TIL/Tc infiltration (Fig. S4a).

2.     Methodsextremely difficult to follow. Even when previously published, methods used in the current manuscript should be at least summarily described. In details:

Line 89: Mantis SCORE for the MSI status definition. Should clarify how used.  

Line 103: unclear how TCR diversity and clonality were calculated in this contest. 

Line 104: SNS (single nucleotide substitution) are subcategory of  SNVsIt is unclear how they were derived. Please specify. According to authors, SNS were determined by fitting somatic single nucleotide variants following the procedure described elsewhere (in their original paper nat comm). I was unable to find this description in the cited ref (21). 

Line 95: According to Authors …All sample had passed tumor purity for inclusion in the original publication. What does it mean? This is important issue for ther MANTIS score More details are needed. 

Line 107: REMARK (for marker assessment) guidelines were used but it’s not specified how.

Line 126: need specifications re how OptimalCutpoints were used

3.     Results: several issues are critical including:

a)     Fig 1A is difficult to interpret without knowledge of the original distribution of   TIL/TC infiltration using a box (or better a violin ) plot that allows to understand how the arbitrary cut-offs used to distinguishing the 3 sub-classes (low, high and very) were chosen.  

b)    The sentence on line 151-154  is criptical (Smilarly, the total single nucleotide substitution (SNS) was not retained as significant, in line with the observations that tumour mutational burden might be of secondary importance compared to immunological factors). Authors defined  SMS (in methods)  according to paper # 21, but in the paper  nel paper SNS are not mentioned.  Presumably – as the name suggests - SNS should be a subset of the mutations, what doest it means that they are not significant?  Please clarify

c)     Are SNS mutations in their genomic contest, i.e. looking at the bases before and after the mutation. To this end it is also unclear whats the difference between SNS and Single Base Substitution (SBS) as defined by  Alexandros 2013, 2014 & 2022). If different from SBS the procedures trough wich SNS  were identified should be reported (from Bam files to the final product).

d)    To the same end it is difficult to interpret fig 3a in light of what said above in c)

e)    The code with wich the TGCA analysis was done should be made available on a open access repository  such as github for example 

f)      On Line 187: the classification "Very High infiltrated cancers" has been made only on the basis of the (>90% percentile) and thus is data-set dependent. Are the cutoffs at the 90% percentile similar of different in TGCA and CTPAC? 

g)     The analysis in fig S5b/c is incompleteIn the analysis of the mutational profiles  according to the cited paper  (line 204, ref 33) the (T >C) sostitution is considered within its  genomico contest. As in comment 3c) authors should report the tricnuclotide genomic contest of each mutation  (Again as done by Alexandros ref 33).

h)    Similarly as per comment 3a above for the TIL/TC distribution, the meaning of  fig 3c is also difficult to interpret  due to the lack of information on the original distribution of the MSI scores  and how they were  calculated with MANTIS. valore. Bisogna vedere ad esempio quanto i best 10 (very high) si differenziano dagli high o dai low. Gli autori DEVONO mostrarci la distribuzione dei valori

i)      In  3.5.1 the authors conclude that …Collectively, our findings show that the "very highly" infiltrated CRCs display distinct genetic features. However no statistical test is provided and thus this ‘conclusion’ remains a merely descriptive  statementEspecially in light of the fact that the "very high infiltration" subclass definition is totally data driven. Without an established cut-off the results could be poorly reproducible and thus not very innovative

Other issues

Acronyms, figures legend and references are riddled by omissions a/o mistakes and should be double checked.

Reviewer 2 Report

   In this article, the authors developed a prognostic model for colorectal cancer, using sequences from TCGA and the CPTAC. The study specifically analyzed transcriptome and TIL/Tc-TCR data to develop the prognostic models and cross-validated the model with the CPTAC data. The study is providing new knowledge and relating the prognosis of the colorectal cancer patients with the infiltration of the lymphocytes whether it is low/high or very high and reporting the genes implicated with each sample and its prognosis. Although more future studies are required to validate the proposed model more, the study provides a starting point and pointing to the importance of linking the lymphocyte infiltration with the patient prognosis. The materials and methods and results section were well-written, as well as the limitations section that highlighted possible caveats and how the study can be expanded on. I think after addressing a few major points, that the paper is informative and sheds light on a new proposed prognostic model for colorectal cancer.

Major points for consideration:

1)    The introduction appears to be concise especially that the study itself includes several genes and pathways, that were not included in the introduction. And highlighting more the importance of the study in comparison to other similar studies, also explanation of why the lymphocyte infiltration was particularly chosen was lacking.

2)    It is not clear which data the authors used in the study, what is the accession number or study number in the used databases of TCGA and the CPTAC? All the relevant used datasets need to be mentioned clearly and preferably in a table.

3)    It is also not clearly found as to the data used from the CPTAC database, how many samples and which ones where they?

4)    How were the 'low or high' groups distinguished from the 'very high' groups? This point needs more elaboration.

Minor points for consideration:

1)    The table for the abbreviations was lacking several terms e.g. HR, SNS, TMB. The manuscript needs to be revised and all the abbreviations included in the table.

2)    In an instant it is referred to as 'sideness' and in other instances it is referred to as 'sidedness', please revise for consistency and for the more accurate term.

3)    Figure 4 wording has a format that is too small to be read. It needs to be with a larger font.

4)    The terms throughout the manuscript need to be proofread for consistency.

Reviewer 3 Report

My dears,

Interesting paper. Please find my comments in the attached file.

General considerations:

The way you constructed the COX model is a bit atypical. You just took variables and inserted them all without previously testing if they have an impact or not. I believe this should be done for all data before constructing a multivar COX.

Tumor or tumour? - choose betwwen american english and british english and then use only that one. don't mix them around
